# A Method for the Validation of Simulated Mixing Characteristics of Two Dynamic Mixers in Single-Screw Extrusion

**DOI:** 10.3390/polym12102234

**Published:** 2020-09-28

**Authors:** Christian Hopmann, Malte Schön, Maximilian Mathias Reul, Martin Facklam

**Affiliations:** Institute for Plastics Processing (IKV) in Industry and Craft at RWTH Aachen University, Seffenter Weg 201, 50274 Aachen, Germany; sekretariat@ikv.rwth-aachen.de (C.H.); maximilian.reul@rwth-aachen.de (M.M.R.); martin.facklam@ikv.rwth-aachen.de (M.F.)

**Keywords:** dispersive mixing, distributive mixing, dynamic mixing elements, flow visualisation, thermal mixing, validation

## Abstract

The field of simulation and optimisation of dynamic mixing elements (‘mixers’) is lacking good methods for spatially resolved validation and flow visualisation. For this reason, the authors present an experimental setup that gives better insight into the thermal, distributive and dispersive mixing process by measuring melt temperatures upstream of the mixer and injecting a secondary, visually distinguishable stream of melt upstream. Running extrusion trials for a polyethylene on both a rhomboidal and a Maddock mixer, temperatures, gray scale distribution of images of extrudates and size of dispersed domains in incompatible polystyrene were measured. It was found that temperatures upstream and downstream of the mixer can be quantified. This was used to validate a simulation of thermal mixing. In distributive mixing, good agreement with simulation and an excellent spatial resolution were observed, thereby identifying an area of the rhomboidal mixer in need of geometric improvement. For dispersive mixing, a trend coherent with extrusion theory was found.

## 1. Introduction

In Section 1, the authors give an overview of single-screw extrusion, the mixing element for single-screw extrusion and simulative optimisation of these mixing elements. They also cover the state of the art for the simulation and validation of single-screw mixers.

Single-screw processes are the most important method of melt plastification in plastics processing. Due to the simplicity in both construction and operation, single-screw extruders enjoy economic advantages compared to e.g., twin-screw or planet roller extruders [1]. Consequently, single-screw extruders are the ‘workhorses’ of most extrusion plants worldwide. There, they serve four crucial roles: first, to draw the bulk material from the hopper into the screw and feed it forward, second, to compress and melt the plastic. The third role is to transport this melt forward against the pressure present at the extruder outlet while the fourth task consists of homogenising melt temperature and equalising the distribution of any other materials (such as colourants) added to the raw plastic feedstock [2]. Within this contribution, these ‘any other materials’ will be referred to as ‘additives’. Conceptually, all four roles are fulfilled by means of a relative motion of the screw relative to the cylinder and the melt.

It should be noted that while the list shown above distinguishes between thermal mixing and the distribution of any other material (distributive mixing), it disregards dispersive mixing. In some situations, dispersive mixing in single-screw extruders is not necessary. Dispersive mixing is—at its simplest—the destruction of aggregates and agglomerates found in either the additive or the feedstock through high forces. Many situations situations in extrusion (such as use of agglomerated mineral additives or the presence gel particles in melt), therefore, require the extruder to also fulfil this role [3].

Two mechanisms contribute to distributive and thermal mixing: diffusion and thermal or material convection [4]. Diffusion of temperature usually is referred to as heat conduction. Considering that plastics are poor conductors of heat [1,2] and very few additives are soluble in plastics melts, convection is the prevalent mechanism [5,6].

However, the complete absence of turbulence in flows of plastic melts leaves only one practical option for this convective mixing: repeated rearrangement of the mass flow. This is usually achieved by splitting the flow into smaller portions, moving them along non-identical paths and finally recombining them. Repetition of these three steps yields increasingly good mixing [7,8]. Some amount of rearrangement is provided by vorticial cross-channel flow during melt pumping in the compression and metering zone of the screw channel [5,9]. This vorticial cross-channel flow can be modified to induce chaotic mixing. During chaotic mixing, the circular, predictable pattern of flow with an unchanged centre of the vortex is broken up. This is usually achieved by some sort of barrier in the screw channel. The role of the barrier is not only to break up the vortex once, but to repeatedly break it up in a different manner each time. [10,11]. Research inspired by in part by the findings of Kim and Kwon [10,11] in this field has led to some crucial findings: Zhu et al. as well as Connely and Kokini observe that twin screw extruders show superior mixing compared to single-screw extruders [12,13]. Zhu et al. also observe even better mixing in triple-screw extruders [14]. Xu et al. confirm another important discovery in finding that distributive mixing performance depends on the initial location of the material that needs to be spread homogenously [15].

In contrast to pumping screw channels are also are dynamic mixing elements (also referred to as mixing sections or dynamic mixers). These are widely used in single-screw extrusion nowadays [1]. In contrast to vorticial cross-channel flow, mixing in dynamic mixing elements offers a far great amount of splitting and recombination per meter of screw length [16,17]. In addition to that, a purpose-built dynamic mixing element can be used to greatly constrict the flow channel. This constriction, usually sized to between 1% and 0.5% of the screw diameter [18], prevents the egress of solid materials (e.g., non-plasticised raw material) and provides good dispersive mixing. It does this by inducing both a high level of shear stress within the flow channel and considerable elongational flow upstream of the constriction. However, when replacing sections of the regular ‘pumping’ screw channel characterised by its high conveying capacity with a dynamic mixing element, the screw becomes less capable transporting melt forward against the pressure present at the screw tip. This obviously reduces possible throughput. Therefore, considerable work has been done on optimizing mixing elements to deliver good mixing capability without strongly affecting extruder throughput. This work has resulted in a bewildering variety of different mixer topologies. Campbell and Spalding list 15 types of mixer in six basic ‘families’ [19], with numerous other types being attested to in other works and patents [20,21,22].

With the advent of modern computer simulation technology, it has been made possible to try out new mixing elements in ‘virtual experiments’ without having to bear the costs involved in the manufacturing of a prototype and the subsequent laboratory trials. As a result, considerable work has been done in this field, covering both simulative design and optimisation as well as evaluation of mixing in extrusion trials and correlation with simulation data. However, as with all simulation, experimental validation is necessary to provide confidence in the simulated data.

Rios et al. use the boundary element method to compare several different designs of rhomboidal mixing sections with regard to pressure loss and residence time distribution, with a subset of the designs also being subjected to laboratory trials including an analysis of colorant distribution and thermal homogeneity [23,24]. However, the simulation is somewhat simplified and thermal mixing is not considered during design and only measured ‘after the fact’.

More recently, Marschik et al. have undertaken a comparable analysis of different block-head mixing screws using the finite element method [25]. Pressure drop, shear heating as well as distributive and dispersive mixing are studied for different geometries, but no validation takes place.

Other recent work by Moritzer and Wittke compares numerous diamond-type, cylinder-type and ‘anticlockwise’ mixing elements in a simulation driven by the finite volume method [26]. Again, both distributive and dispersive mixing are considered, but thermal mixing is neglected. No comparison to laboratory trials is made.

Gorcyzca analyses both distributive and dispersive elements for high-speed extruders. Simulative and laboratory trials are conducted [27]. Using infrared thermography to determine melt homogeneity at the outlet, he observes good qualitative agreement after manually adjusting thermal boundary conditions in the simulation. The necessity of these arbitrary adjustments makes it difficult to employ simulative optimisation in situations in which no roughly similar mixing element geometry is available for reverse engineering. Dispersive mixing is not analysed during the laboratory trials, while the simulation data for dispersive mixing is found to be difficult to interpret. The analysis of distributive mixing in experimental trials is done by comparing average colors and subjective judgement. It should be noted that Gorcyzca makes an effort to determine this mixing not only downstream of the mixing elements, but also upstream.

Neubrech et al. also apply varying temperature profiles to the mixing element inlet across multiple simulations until good agreement with experimental data is found [28]. A valid temperature profile then is used during a simulative comparison of two different dynamic mixing rings. The numerically determined reduction in pressure drop is validated in experiments, while the reduction in shear heating leading to lower melt temperatures at the outlet is not. Neubrech et al. subjectively rate blown film samples in terms of visual homogeneity. This homogeneity is taken as a measure of mixing performance. However, visual film homogeneity is not correlated to simulation data.

Four different Maddock-style mixers are studied by Sun et al. [18,29]. Their focus lies on pressure drop, distributive mixing, residence time distribution and melt temperature. Experimental validation only covers pressure drop and a rudimentary look at melt temperature.

Perdikoulias and Kikuzawa also consider variations of a Maddock-type mixer in their simulations [30]. The work of Potente and Többen [31] as well as the works of Wang and Tsay [32] goes in a very similar direction. Without considering experimental validation, they focus on determining the correct sizing and orientation of the shear gap. Comparable work is done by Pandey and Maia [33] or Rauwendaal et al. [34] who use numerical flow simulation to assess the rate of elongation produced by novel types of mixing element. Another approach to this purely simulation-based method is presented by Janßen and Schiffers, who intend to automate mixing element design [22].

Experimental validations of finite element calculations on Maddock-type mixers using grayscale values determined by an inline melt camera are conducted by Kubik et al. [35].

Multiple researchers from IKT in Stuttgart (most notably Celik, Erb and Bonten) are known to work on several fields related to the design of mixing elements: They use simulation technology to analyse the influence of rheological behaviour on mixing [36]. Additionally, they investigate varying methods to determine criteria for the both interpretation of simulation [37,38] and extrudate photography data [39]. In a third approach, they have established a novel machine setup to measure dispersive mixing by means of combining incompatible plastics [40].

When viewing the field of simulative design of mixing elements and the validation of these simulations, the following gaps in the state of the art can be identified (see also Figure 1):

Validation of simulation experiments on dynamic mixing elements is relatively rare. Of the 15 simulation-driven research works cited as [23,24,25,26,27,28,29,30,31,32,33,34,35,36,37,38,39], only three [27,28,39] validate their findings. When validation is done, it frequently takes the form of only comparing the simulated and real pressure drop across the entire mixer, e.g., in [24,27]. Frequently, validation is only undertaken by comparing extrudate properties such as temperature and colorant distribution as a function of mixing element geometry and operating point, e.g., in [24,27]. Neither method allows a highly resolved analysis of local flow velocities and material temperatures. It lacks a good spatial resolution. This is because only an integral value of mixing (e.g., the standard deviation of the gray scale values) is determined. The integral frequently subsumes either the entire mixing element or even the entire screw. In the case of distributive mixing, great difficulty is encountered when trying to condense the information of extremely noisy grayscale images of extrudates into a single value. The high level of noise is partially caused by the high amount of mixing depicted.

In view of this situation, the authors’ contribution evaluates a method that allows researchers to validate simulations of thermal mixing, distributive mixing and dispersive mixing with a higher level of confidence. The method achieves this by adding a robust temperature sensor upstream of the mixer and enforcing the state of distributive and dispersive mixing in this location by injecting a well-defined secondary stream of melt. Under these conditions, simulation data that exists at a high spatial (in this case: radial) resolution can be correlated to similarly spatially resolved experimental data, thereby providing a great level of confidence. In conclusion, the hypothesised method has a far greater spatial resolution and, therefore, should make it possible to correlate simulation and experiment data at single operating point. It should also remove the necessity to analyse mixers at several operating points in simulation and experiment and correlate the changes between the operating points in simulation and experiment.

Accordingly, the following hypothesis is formulated for this contribution:
The proposed method offers a radial resolution of thermal, distributive and dispersive mixing in experiments. This resolution is closer to the high radial resolution of simulation data than the previous state of the art.

In the following segments, the authors will attempt to validate this hypothesis by means of comparing simulation data to results from with both ‘classical’ and spatially resolved interpretations.

## 2. Materials and Methods

In Section 2, the authors present the material and methods used during the extrusion trials (Section 2.1, Section 2.2, Section 2.3) and the simulative recreating thereof (Section 2.4).

### 2.1. Materials

In this section, the authors describe the raw materials used and their properties.

Laboratory extrusion trials were carried out with Hostalen GD 9550F (LyondellBasell GmbH, Wesseling, Germany), a high-density polyethylene (HDPE) typically used for blown film applications as the primary material [41].

Two different materials were employed as a secondary melt stream: Firstly, a mix of Hostalen GD9550F and 5% (by weight) of the black colour masterbatch Polyblak 7392, LyondellBasell GmbH, Kerpen, Germany) and secondly, polystyrene (PS) 156F (INEOS Styrolution Group GmbH, Frankfurt am Main, Germany) also coloured black. The black colour was achieved by adding 20% by weight of carbon black (Spheron 6400A, Cabot Corporation, Alpharetta, GA, USA). Compounding of this material took place on a ZSK 26 twin screw extruder (Coperion GmbH, Stuttgart, Germany). While the first material was expected to blend with the primary (or main) flow of melt, the polystyrene was incompatible with the polyethylene of the primary flow. Therefore, it was expected to form a separate domain. As described in [40,42] shear and elongational loading reduces this domain into smaller domains, with elongational stresses being more effective. Consequently, the size of polystyrene domains is an excellent indicator of dispersive mixing. Figure 2a shows the rheological data of both Hostalen GD 9550 F and the carbon-filled PS 156F, which can also be described by the Carreau model [43] in Equation (1) with the parameters provided in Table 1. In the Carreau equation, A describes the zero-shear viscosity, B the reciprocal transition rate in and C the slope of the viscosity curve. α_T_ is the shift factor for temperature and can be calculated from the Williams-Landel-Ferry equation in Equation (2) [44].
(1)$$\eta =\frac{\alpha_{T}*A}{{\left(\alpha_{T}*B*\dot{\gamma }\right)}^{C}}$$
(2)$$\log\left(\alpha_{T}\right)=\frac{-C_{1}*\left(T-T_{r}\right)}{C_{2}+\left(T-T_{r}\right)}$$

In this equation, *C*_1_ is 8.86 and *C*_2_ is 101.6 K, while *T_r_* has value of 237.25 K for HDPE and a value of 341.06 K for PS.

For shear rates between 0.01 and 1000 1/s, the ratio of viscosities as calculated from the Carreau-Williams-Landel-Ferry models can be described by a power law (see Figure 2b). For 0.01 1/s, the ratio is ca. 4.4 to 1, for 45 1/s it is 3 to 1 and for 1000 1/s the ratio is 2.7 to 1, with the HDPE being the more viscous fraction. At this ratio of viscosity, breakup of drops of the dispersed phase is possible by both shear and elongation according to the theory of critical capillary numbers put forward by Grace [45]. However, elongation is expected to perform better [45].

### 2.2. Laboratory Extrusion Trials

For the extrusion trials, the authors used a single-screw extruder (type 6E4/27D) produced by Oerlikon Barmag GmbH and Co. KG (Remscheid, Germany), Remscheid. The extruder’s cylinder has a grooved and water-cooled feed zone while its diameter *D* is 60 mm and its total length is 27 *D*. The screw utilised was a standard three zone design (no barrier) with each zone having a length of 8 *D*.

However, instead of using the remaining cylinder length of 3 *D* for the dynamic mixing element, the screw was extended forward by means of an adapter of reduced diameter. By choosing the length of this adapter to be 3 *D*, the beginning of the mixing element was placed just outside of the main cylinder. Finally, a cylinder extension was mounted to the extruder cylinder, covering the dynamic mixer.

This test setup made it possible to determine the radial temperature profile upstream of the mixing element. Immersed multipoints temperature sensors manufactured by TC Mess-und Regeltechnik GmbH (Mönchengladbach, Germany) were used for this purpose. They were made up of three type J thermocouples located at a distance of 2.5 mm from each other, as seen in Figure 3 and marked with an 8. A metal cylinder of 6 mm diameter filled with mineral insulation surrounded the thermocouples and protects them against mechanical deformation. Additionally, melt pressure was measured at this location using a strain gauge-based pressure transducer produced by KMK Sensortechnik GmbH (Schorndorf, Germany). Both sensors were mounted in a half-inch 20 UNF 2A-type tapped hole with conical sealing surfaces. A third hole of this type was also located upstream of the mixing element, feeding a secondary stream of melt to the mixer inlet. This hole had a diameter of 6 mm and terminates flush with inside of the cylinder extension. The secondary melt was added as an ‘outer layer’ at maximum radius of 30 mm. A Polytetrafluoroethylene/braided metal heating hose was used to make a connection between the third hole and the secondary extruder, a smooth-bore 19/25 D Extrusiograph plasticating unit on an Plastograph EC (*D* = 19 mm, Length of 25 *D*; Brabender GmbH and Co. KG, Duisburg, Germany), which served to plasticise and pressurise the secondary melt stream. Further information concerning the screw designs can be taken from Table 1. Figure 3 and Figure 4 show the entire experimental setup both schematically and in a photograph.

Conceptually, this addition of visually distinguishable and/or incompatible secondary material upstream of the mixer reproduces the works of numerous other researchers, such as Celik and Bonten [40] or van der Hoeven et al. [46] and might also be compared to the use of sample collectors ahead of the mixer as undertaken by Gorczyca or Carson et al. [27,42]. However, using a secondary extruder with comparatively high throughputs made it possible to analyse distributive mixing across the entirety of the extrudate cross section easily. This was the result of a high contrast, which in turn was caused by a lack of dilution of the secondary melt. This lack of dilution followed from a high volumetric percentage (~2.5% or 1 to 40) of the secondary melt. In addition to that, the secondary extruder was a very simple solution to the challenge of having to plasticise the secondary melt and inject it against high pressure.

The end of the cylinder extension coincided with the tip of the mixing element. Attached to the cylinder extension was a flange with an adjustable fingertip-type flow restrictor. A pressure transducer and multipoint temperature sensor setup, temperatures and pressure at the mixing element outlet were also used to measure melt data at the inlet of the flange. The difference at this measuring location lay in the fact that the thermocouples were spaced apart at 7 mm intervals here. The flow restrictor served to simulate the pressure loss of an extrusion die downstream, but was also used to ensure identical pressures (and thereby, melt properties) at the measurement points regardless of mixer used or throughput.

Over the course of the extrusion trials, two different mixing elements were used. The first was a Maddock-type mixing element with 4 shearing flights (or shearing gaps) sized at 0.6 mm and a pair of channels leading to and from that gap. These channels possessed a triangular cross section with a maximum depth of 6.5 mm. The shearing flights were terminated circumferentially by 4 wiping flights with a clearance of ca. 0.25 mm between flight tip and cylinder. The second mixing element was a rhomboidal mixing element with 49 ‘diamonds’ arranged in 4 equidistant flights with a pitch of 3 *D* each. Each ‘diamond’ measured 7.5 by 8 by 6 mm. Figure 5 shows photographs and unwound schematic depictions of both mixing elements.

Each mixing element took part in four extrusion trials: they were analysed at throughputs of 20 and 80 kg/h each, while for each throughput one trial with polyethylene plus black masterbatch as secondary material and another trial with polystyrene as a secondary material were conducted. At each operating point, pressures and temperatures were measured for 20 min after reaching a steady state. Zone-dependent cylinder temperatures and observed revolutions per minute can be seen in Table 2 and Table 3. Based on past experience with the 19 mm extruders throughput behaviour at IKV [47,48], the secondary extruder rpm was set to ensure a roughly 40:1 ratio between primary melt and secondary melt throughput. When processing PS on the secondary extruder, the feed zone temperature was increased to 45 °C in order to achieve higher friction between the PS pellets and the cylinder, resulting in a much improved solids transport which in turn ensured the 40:1 throughput ratio.

### 2.3. Extrudate Sample Analysis Methods

In this section, the authors describe the methods using during extrudate sample analysis.

Extrudate samples of ca. 200 mm length were taken every three minutes. As each extrusion trial was run for 30 min, a total of 10 samples were collected. In order to prevent the formation of shrinkage cavities, the material was collected and allowed to cool without any intervention. Afterwards, segments were produced by cutting the samples normal to the machine direction. One segment per extrudate sample was ground (grain sizes 80 and 600) and photographed with a camera of the type Alpha 6000 (Sony Europe B.V., Hoofddorp, The Netherlands) [Settings: F8.0, 1/10s, ISO 100, 30 mm]. Samples taken during extrusion with polystyrene as secondary material were expected to contain discrete domains of polystyrene incompatible with the polyethylene matrix and were, therefore, analysed further.

Using a RM2265 microtome (Leica Biosystems GmbH, Nussloch, Germany) sections with a thickness of 25 µm were prepared and then viewed with a digital microscope (VHX5000, Keyence Deutschland GmbH, Neu-Isenburg, Germany), applying a bright-field technique. Images of both segments and sections are analysed using ImageJ (National Institutes of Health, Bethesda, MD, USA).

When analysing the segments, the images were binarised (turned into a grayscale image), with each pixel being assigned a brightness value between 0 and 255. The standard deviation of this brightness value across a single image is referred to as δ_gray_. High values of δ_gray_ correspond to a great variety in color and thereby to poor mixing. This method has been used numerous times in analysis of extrudate quality [5,27,35], but was also applied to images exported from the flow simulation in this work. In the photographs taken by the camera mentioned above pixels were sized to ca. 30 µm. Therefore, grayscale values provided an integral, indirect value of colorant distribution for a square of 30 µm by 30 µm.

During analysis of the sections containing polystyrene domains, ImageJ was used to first identify the borders of each domain and then the area of each domain. Histograms of domain area were generated for each image. The classes of the histograms covered 200 µm^2^ each, while for each class both the relative and cumulative percentage was calculated. These percentages were relative to the total area covered by polystyrene domains.

### 2.4. Simulative Recreation of the Extrusion Trials

In this section, the authors describe how they used a modern simulation software to virtually recreate the extrusion trials.

Flow simulation using the finite volume method in OpenFOAM (OpenFOAM Foundation Ltd., London, UK) also took place. The SIMPLE algorithm [40] was implemented in OpenFOAM. In the modified version, the full incompressible Navier–Stokes equations for volume conservation and impulse conservation were solved. The Appendix A give the numerical settings in more detail. The equation of energy conservation was modified to become a temperature equation that depicted shear heating (also called viscous dissipation) [49,50]. The authors also implemented a Carreau-Williams-Landel-Ferry type model [43,44] for temperature- and shear-rate-dependent viscosity [49]. In order to map the rotatory relative motion between screw and cylinder in a static computational grid, a multiple reference frames (MRF) approach was chosen [51]. In one area a fixed reference system was used, in the other area a rotating reference system was used. The latter area was presumed to cover the entire mixing element except for the cylinder boundary patch. Consequently, the mixing element could be modelled by a static hexahedrally-dominant mesh. Within the MRF zone, an additional velocity was applied, which is as calculated in [51]:**u**_system1_ = **u**_system2_ + Ω**L**(3)

Herein, **u** describes the vector of the flow velocities at any given cell, Ω the scalar angular velocity and **L** the vector pointing from the center of the rotation axis to the respective cell center. Figure 6 shows a graphical view of how system 1 and system 2 are located in the case of a Maddock mixer simulation.

The system of equations described above utilises simplifications typical for extrusion processes as described by Hopmann and Michaeli [52]: It only considers steady-state laminar flow, neglects body forces (i.e., gravity), assumes no influence of pressure on melt viscosity and a constant melt density of 736 kg/m^3^. Hopmann and Michaeli also consider perfect adhesion (‘no-slip condition’) to the wall realistic in most cases. Nevertheless, wall slip in extrusion of high-density polyethylenes has been detected by several other researchers [53,54,55] and found to be compatible with the models of the molecular and atomistic structure of high-density polyethylene [56,57]. However, the work of Hatzikiriakos find ‘that melt slip occurs at a critical shear stress of approximately 0.09 MPa’ [53]. In the authors’ simulations, wall shear stress across all simulations reached a maximum ca. 0.06 MPa at the boundary. In view of this, a no slip condition at the wall was assumed as realistic modelling choice.

Both mixer geometries were meshed with the snappyHexMesh tool provided with OpenFOAM. The authors employed a hexahedral mesh with a base resolution of 80 cells in both radial directions and 160 cells in the axial direction. This mesh was refined in all three axes to have double that resolution for radii between 24.5 mm and 28 mm and quadruple the base resolutions for radii between 28 mm and 30 mm. By means of this, the gaps between the tops of the shearing and the wiping flights and the cylinder could be depicted in the simulation. Likewise, the gap between the top of the rhomboidal elements and the cylinder was depicted. Figure 7 and Figure 8 show the meshes in more detail, while the Appendix A that feature a general overview of the meshes.

Four different simulations were realised: For both the Maddock and the Rhomboid mixing element throughputs of 20 and 80 kg/h were simulated. A Dirichlet boundary conditions for the inlet was investigated. Here, a radial inlet temperature profile derived from the multipoint temperature measurements taken during the extrusion trials was used. Other thermal boundary conditions were as follows: Neumann boundary condition (*δT/δ**n*** = 0) for the mixing element and outlet and Dirichlet boundary condition (*T* = 483.15 K) for the cylinder. In the Neumann boundary condition, **n** signifies the surface normal. Accordingly, the thermal Neumann boundary condition prescribes that for every boundary face of the mesh, the gradient of *T* normal to the surface of that face is zero. By enforcing the Neumann boundary condition with a zero gradient for the screw surface, no heat transfer between the melt and the screw is modelled. Expressed in other words, this means that the melt coming close the screw surface will heat up the screw surface until a thermal equilibrium is reached. The authors assumed that at a steady state, this equilibrium temperature was affected only by heat transfer processes in the melt and that heat transfer processes in the screw could be neglected. Two reasons support this assumption: firstly, both mixing elements were periodic with regards to the extrusion axis. Heat transfer in the radial direction was expected to be negligible. Secondly, shear heating occurred somewhat evenly along the extrusion axis. This would lead to an axial temperature gradient. The authors expected the heat conduction caused by this gradient to be negligible compared to the convective heat transfer. For a throughput of 20 kg/h, a mixer length of 180 mm, a melt heat capacity of 2100 J kg^−1^ K^−1^, a screw cross section of 0.001 m^2^ and a screw thermal conductivity of 60 W·m^−1^·K^−1^ heat transfer by convection was ~12 Watt per degree Kelvin of the axial temperature difference. In the same scenario, heat conduction accounted for ~0.25 Watt per Kelvin. For 80 kg/h, the numbers are ~48 Watt per Kelvin and ~0.25 Watt per Kelvin. Heat conduction in the screw also was neglected by [23,24,25,26,27,28,29,30,31,34,35,36].

The boundary conditions are also summed up in Figure 6. The throughputs mentioned above were prescribed at the inlet as a fixed volume stream. Finally, angular velocity was calculated from the RPM observed in the experiments.

The resulting simulated flow field was somewhat different depending on the type of mixer considered. Figure 9 shows a visualisation of flow velocities in the Maddock mixer, while Figure 10 shows the same for the rhomboidal mixer. Both simulations were conducted at a throughput of 20 kg/h. A more complete flow field visualization is available in the Appendix A as Appendix A.

As seen in Figure 9, the flow in the Maddock mixer is dominated by the rotation, while a most important other direction is the flow along the extruder axis. A combination of these two makes it likely that any volume of melt passes over the shearing or wiping gap at least once. The strong rotational character also affects the melt in the channels leading to or away from the shearing gap. The melt here can be understood to travel along with the rotation of the mixer, with the motive force towards the outlet of the mixer only provided by the pressure upstream of the mixer. This causes a considerable amount of flow in the radial direction.

In stark contrast to this, there is almost no radial flow in the rhomboidal mixer. Instead, the rotational flow is combined with splitting motion ahead of each rhomoboidal ‘diamond’. Flow velocity is highest close the ‘diamond’. In addition to that, high velocity flow occurs in the gap between the ‘diamond’ and the cylinder. Again, rotational flow is far stronger than flow along the extruder axis. Compared to the Maddock mixer, axial flow is slower. This can easily be explained by the fact that the cross section of the flow channel is far smaller in the Maddock mixer. Since both mixers are operating at the same throughput, the axial flow rates in smaller cross section must be higher.

## 3. Results

In this section, the authors compare simulation data with two different validation approaches: On the one hand, a conventional validation method based on single point data and analysis of changes brought by adjustments of geometry and the operating point, on the other hand a spatially resolved method.

This comparison is done for thermal mixing in Section 3.1, for distributive mixing in Section 3.2 and dispersive mixing in Section 3.2.

For brevity, the Maddock mixer is referred to as MM, while the rhomboidal mixer is referred to as RM. Similarly, results from experiments conducted with polyethylene as secondary material are denoted by ‘-PE’, while results from experiments conducted with polystyrene as secondary material can be recognised by the ‘-PS’ suffix. It must be noted that owing to the difficulty of describing two-phase flow, the simulations only aimed to recreate the extrusion trials in which a polyethylene secondary melt was used.

### 3.1. Spatial Resolution of Thermal Mixing

In the first step, experimental and simulative data concerning thermal mixing/radial temperature profiles was analysed to determine whether the authors’ hypothesis held true for thermal mixing. Figure 11 and Figure 12 show temperature measurements taken upstream of the mixing element for throughputs of 20 kg/h and 80 kg/h respectively. Due to the fact the immersed sensor must be welded close at its tip and a layer of insulation between the thermocouples and the metal body must be added, no temperatures can be measured at the radii between 25 mm and below. In order to still generate data for the entire inlet of the flow simulation, the four temperature profiles for 20 kg/h and the four temperature profiles for 80 kg/h were each averaged and a continuing linear rise in temperature towards the screw was assumed. The radial area affected by this assumption is marked by the gray overlay.

The abscissa expresses the radius, with 0 mm being the centre of the extruder axis, 19.5 mm the maximum diameter of the adapter and 30 mm the cylinder.

Even though neither the choice of mixing element nor choice of secondary melt should affect the inlet melt temperature profile for a fixed throughput, the absolute temperatures do not reflect this. While all temperature profiles show a linear gradient to temperatures rising toward the screw, there is considerable overlap. For 20 kg/h the plots for experiments conducted with polystyrene as secondary material overlap, while the plots for experiments conducted with polyethylene overlap at somewhat higher temperatures. For 80 kg/h, the plots associated with the Maddock mixing element/rhomboidal mixing element overlap, with the former showing higher temperatures. This indicates that another factor varying from experiment to experiment is most likely causing this effect. As the experimental setup required both disconnection and removal of the upstream immersed temperature sensor and disassembly of cylinder extension and flange, it seems plausible that variations in sensor position and electric connection occur. This interpretation is supported by the fact temperatures at the cylinder change between measurements.

Nevertheless, when viewing the data per plot, a consistent result can be observed: For 20 kg/h, temperature rises nearly linearly by approx. 3 °C between the cylinder and a radius of 25 mm, for 80 kg/h the nearly linear temperature across the same distance is 9 °C. This observation matches well with the expected increase in shear heating at higher rotational screw speeds and likewise expected reduction in residence times (and consequently, reduction of thermal homogenisation due to heat conduction).

Generally speaking, the radial temperature profile in single-screw extrusion is affected by viscous dissipation (‘shear heating’) on the one hand and heat exchange with the barrel on the other hand. Previous research [58,59,60] has established that the highest temperatures will be found at the screw root or at the extrusion axis. This is the consequence of a) the high amount of shearing and shear heating present between the moving screw and the viscous melt and b) the fact that the screw, unlike the barrel, cannot effectively release heat to the environment [60]. The screw cannot do that because it is surrounded by plastics with low heat conductivity in the radial direction. The axial connection with gearbox and motor has a small cross section compared to the overall screw surface [1]. The radial temperature profile between the screw and the cylinder either shows a linear/parabolic drop towards the barrel temperature or—if the barrel temperature is high—a U shape, with a local minimum of temperature roughly located halfway between screw and cylinder [59]. For high viscosity HDPE at low temperatures, Abeykoon et al. observed maximum melt temperatures exceeding the cylinder temperature by 15 (for low screw speeds) to 25 °C (for high screw speeds) [58].

Our work confirms those findings, with increasing screw speeds resulting in a higher melt temperature overall, with the maximum of the radial temperature profile located at the screw. This is caused by the fact that shearing and shear heating are most prevalent close to the screw and the fact that, unlike the barrel, the screw cannot effectively release heat to the environment.

In addition to temperatures upstream of the mixing element, temperatures downstream of the mixer have been measured as well and are presented in Figure 13 and Figure 14.

Here, a more coherent pattern can be observed: independent of the mass throughput, the utilisation of the Maddock mixing element always results in higher temperatures compared to the rhomboidal mixing element. The addition of polystyrene as a secondary melt leads to a reduction in temperature rise. Finally, at 80 kg/h, the rise in temperature is far higher compared to the operating point of 20 kg/h. The entirety of these findings agrees with previous knowledge laid out in e.g., [46]. The shear-intensive Maddock mixing element generates more viscous dissipation, especially at high rotational speeds. Also, the addition of low-viscosity polystyrene reduces shear heating.

The authors’ results shown in Figure 13 and Figure 14 are consistent with the finding of e.g., Abeykoon et al. or Kelly et al. [58,60]. Again, for a high-viscosity HPDE the maximum of melt temperature is observed in the centre, with higher maximum temperatures for higher screw speeds. The temperatures at the outlet exceed the temperatures at the inlet, which is caused by shear heating. As the Maddock mixer induces more shear, the maximum temperature is found when operating the Maddock mixer at a high screw speed. Conversely, the rhomboidal mixer causes less shear and leads to lower temperatures. The addition of a low-viscosity polystyrene as a secondary material leads to a drop in temperature increase. As the other process parameters stay constant, a reduction in shear heating must be the root cause. The authors assume that the low-viscosity polystyrene works like a lubricating film, reducing the amount of heat produced during the same movements.

The spatial resolution of the thermocouple measurement locations at the outlet does leave gaps. However, the presence of ‘high’ temperatures at the 10 mm measurement location gives the authors reasonable confidence that no ‘U-shaped’ temperature profile was present at the outlet during the extrusion trials. If such a ‘U-shaped’ temperature had been present, a steep drop of temperature with rising radius would have been observed.

Finally, downstream temperatures, determined by laboratory experiments, were compared to profiles derived from simulation data in Figure 15 and Figure 16. A more complete view of simulated melt temperatures is presented in the Appendix A. It must be noted that the corresponding simulation used the temperature profiles from Figure 11 and Figure 12 as boundary conditions on their inlets. Since the simulation did not cover the effect of adding a secondary material with greatly varying rheological parameters, only experimental data gained from results with polyethylene as secondary material were used for comparison.

The simulation delivered an acceptable prediction of the overall shape of the temperature profile in most cases, except for the rhomboidal mixer at a throughput of 80 kg/h. Absolute temperatures were generally underpredicted, especially for the operating point of 80 kg/h. The general relationship of a great temperature rise in the Maddock mixer and increased shear dissipation at higher rotational speed was qualitatively represented by the simulation. However, for 20 kg/h the simulated temperatures for the rhomboidal mixer exceeded those of the Maddock mixer. Considering the poorer quality of the rhomboidal mixer simulations (which was visible from the large absolute temperature difference present at 80 kg/h), some conceptual error in the RM simulation might have been at fault.

In conclusion, the spatially resolved measurement of melt temperatures before and after the mixing element yielded reasonable results. The temperature profiles agreed with literature data such as the works of Kelly et al. or Schöppner and Resonnek [60,61]. In validation, spatially resolving melt temperatures did offer some advantages compared to typical validation approaches which usually employ a single thermocouple at a depth of a few milometers. For the geometries, operating points and the material analysed here, such a simple measurement method would have been able to detect the significant difference in melt temperature. It would, however, not have been able to deliver a good judgement on whether the overall shape of the temperature profile is predicted correctly.

### 3.2. Spatial Resolution of Distributive Mixing

In this section, experimental and simulative data for distributive mixing are analysed to determine whether the authors’ hypothesis hold true.

In order to predict the path of the secondary melt through the extruder, the convective-diffusive transport equation for a passive scalar *c* is solved, following the example of e.g., *Kerstein* [62]. In this setup, a steady-state differential advection-diffusion equation is solved numerically:(4)∇∗(u∗*c)+∇∗(DC∗∇∗c)=0

Again, **u** describes the flow velocity vector. DC is the coefficient of diffusion, chosen to be 10^−12^ in order to achieve good numerical stability. This approach was used instead of conventional particle tracking to avoid the loss of particles. During the extrusion trials, the rotating screw moved relative to the injection port which was fixed to the cylinder injection. As the port was located 15 mm upstream of the mixer, the authors assume that the secondary melt stream would transform into a continuous drawn-out area at the edge of the radius before entering the mixer. This continuous area was modelled in the simulation as well. In the simulation, the mixer was fixed in place, just like area described above. The location of this area relative to the position of the mixer was expected to impact the mixing performance. However, as no better numerical method was available to the authors, this approach was chosen.

To recreate the transformed area, a Dirichlet boundary conditions was applied to the inlet, with a continuous area at the outer edge of the radius providing *c* = 255 (thereby appearing as black) while the remaining inlet area provided *c* = 0, thus appearing white. The continuous area was sized to cover approx. 1/40 of the inlet. This was done because it was not possible enforce providing *c* = 255 (appearing as black) on an area of very little radial distance. If the area has very little radial distance, numerical errors make it impossible to successfully approximate a solution of Equation (4). Figure 17 shows the inlet boundary condition graphically. It also show the location and distances of the injection port and the measurement plane.

Equation 4 describes a steady-state result without any dependence on time. This is meant to recreate the laboratory extrusion trials in which both the main and secondary extruder were in constant operation and extrudate samples were only taken after a steady-state operating point had been reached. Mixing problems in real production extrusion lines will likely show some dependence on time and therefore require an analysis of ‘back-mixing’ capabilities. However, time-dependent mixing problems are not considered in this work.

In reality, the primary melt without black colourant and the secondary melt coloured black will vary in their thermo-rheological properties. However, as the weight of the colourant only was 0.125% of the total mass extruded, these changes were neglected in the simulation. It must be noted that owing to the difficulty of describing two-phase flow, the simulations only aim to recreate the extrusion trials in which a polyethylene secondary melt was used.

In this case, the melt properties were assumed to be identical for the purposes of the simulation. Consequently, there was no ‘feedback’ from the colourant distribution the pressure, viscosity and/or velocity fields. These were assumed to be static for the purposes of Equation (4).

For evaluation of the simulated mixing, the outlet/measurement plane of the mixer was exported as a *.png image. In the *.png color space, the highest local concentration *c* at the outlet was given an intensity value of 0 (pure black), the lowest a value of 255 (pure white). The images were then analysed δ_gray_ using ImageJ (National Institutes of Health, Bethesda, MD, USA) as if they were photographs of extrudates.

Figure 18 depicts the results for the laboratory extrusion trials with both polyethylene and polystyrene used as secondary melts. Since the results are almost identical, further images will only show results for polyethylene.

Figure 19 depicts the results of extrusion trials compared to simulated concentration fields. It must be noted that owing to the difficulty of describing two-phase flow, the simulations only aim to recreate the extrusion trials in which a polyethylene secondary melt was used. A more in-detail view of simulated concentration fields is available in the Appendix A.

The first observation is the complete lack of influence of throughput on either simulated or measured mixing. However, this is to be expected since the volumetric ratios between primary and secondary melt remain the same. When also considering the negligible influence of inertia on the flow velocities and the fact that both melts show the same shear-thinning characteristics, this observation agrees with fluid dynamics of polymer extrusion.

A comparison of the Maddock mixer and the rhomboidal mixer shows that in both experiment and simulation the former achieved better mixing. The advantage of the Maddock mixer is both visible in the values of δ_gray_ and the spatial distribution of colours. In extrudate samples, that have only passed through the rhomboidal mixer, the outside border is relatively dark and the centre fairly bright. The intermediary area shows a somewhat continuous gradient between the extremes, with several fine lines of black interspersed. A similar observation can be made for the samples that passed through the Maddock mixer. However, in the latter case, the outside border was brighter and the bright centre was interrupted by a darker spot. Images derived from the simulation show the very same details, including the brighter border and the dark spot (see Figure 19).

This pattern can be explained by the fact that secondary melt enters the system from the outside. The ‘teeth’ of the rhomboidal mixer do not force a movement of the melt normal to the extrusion axis, instead only hastening or delay the axial movement of the secondary melt. In contrast, the Maddock mixer forces reorientation by having wiper flights in almost direct contact with the cylinder. Some melt is ‘scraped off’ by these flights and forced down, while other material is forced upward as it passes through the triangle-shaped channel before passing through the shear gap.

This rearrangement of melt could be seen in both simulation and experiment, even though the absolute values of δ_gray_ exhibited different levels. While the simulation data were turned into images by a finely controllable algorithm, the brightness of images of extrudate samples affected the lighting conditions and camera setting. In theory, both algorithm and lightning conditions/camera settings could be tuned to achieve an identical level of δ_gray_ in both simulation and experiment by compressing or stretching out the spectrum of gray values present.

When observing the respective changes in δ_gray_ caused by ‘changing’ the mixing element, it became apparent that this change took a value of ~19 in all four scenarios. This again underpins the observations that the absolute value of δ_gray_ was meaningless, since it could be ‘dialled in’. The stability of the standard deviation value ~19 also indicated a high level of simulation accuracy at both low and high throughputs/screw speeds.

While the change in values of δ_gray_ between Maddock and rhomboidal mixer showed excellent agreement between laboratory experiment and simulation, there was still a visible difference when comparing the images. On the one hand, the simulation did not resolve the fine lines present in the real extrudate samples. This was because these lines were smaller than the grid of the mesh. Therefore, they could not be resolved with Equation (4). On the other hand, the simulation results in images in which the centre of the mixing process were identical to the centre of the sample. The extrudate samples, this center of mixing was shifted somewhat. This shift was caused by two factors: firstly, the melt passed through an asymmetrical flow restrictor downstream of the mixer and secondly, the shape of the extrudate was—as already explained above—not frozen after exiting the machines. Instead, the material was allowed to cool without any intervention. During this, the extrudate changed in shape due to gravity, causing a widening of the ‘bottom’ and a thinning of the top. This movement also contributes to the shift.

When considering the hypothesis advanced by the authors (the question of the usefulness of a high spatial resolution in validation of simulated distributive mixing), two remarks must be made: Firstly, the higher degree of resolution shown here is introduced by the experimental setup and not by changes to the simulation or the method of extrudate analysis. Secondly, even though the scalar/cumulative value of δ_gray_ already proves that mixing is better in the Maddock mixer, the high resolution of the experimental observation enables a clear interpretation of just how this change in δ_gray_ is caused. Therefore, it has been revealed how the rhomboidal mixer geometry will have to be adjusted in order to improve mixing normal to the direction of extrusion. This modification will have to be some manner of ‘wiper’ that forces melt in close proximity of the cylinder towards the screw ground.

### 3.3. Spatial Resolution of Dispersive Mixing

In this section, experimental for dispersive mixing is analysed to determine whether the authors’ method can or cannot be used to give reliable information on dispersive mixing.

Dispersive mixing can also be studied by analysing polystyrene domain distribution at numerous locations. Due to the large amount of work involved in sample preparation for bright field microscopy, the authors will only demonstrate the general suitability of their approach for assessment of dispersive mixing. To this end, microtome sections extracted from the same ‘dead center’ location on extrudate samples are compared for 20 kg/h and 80 kg/h and both the Maddock and rhomboidal mixer.

Consequently, no comparison with spatially resolved simulation data can be made. Figure 20 shows that the location from which the microtome section has been extracted is the centre of the sample. The extrudate sample itself is ca. 40 mm across and the microtome section is ca. 4 mm across. Figure 20 also shows the method used to determine the size of polystyrene domain and the methods used to display the data. Panels (a) and (e) show the samples and the location from which the microtome section has been taken, with panel (a) referring the Maddock mixer and panel (e) referring to the rhomboidal mixer. The microscope images are shown in panel (b) and (f), respectively. Black specks in the microscope images are the polystyrene domains. Finally, panels (c) and (g) show the microscope images after digital processing. By creating a binary picture, the size and area of polystyrene domains can be easily and reproducibly determined. Again, (c) refers to the Maddock mixer and (g) to the rhomboidal mixer. To the left of the image sequences, histograms (d) and (h) with a bin (or class) step width of 200 µm^2^ are shown. These depict the distribution of polystyrene domains. The green line indicates how much of total polystyrene area is taken up by the current bin, while the black line shows a cumulative tally.

Figure 21 shows the cumulative percentage (i.e., the total area covered by polystyrene domains in relation to the area covered by the current class of the histogram and all small classes) in relation to the size of the current class.

A steep increase at classes of low size indicates the presence of many small particles. While the same general shape is recovered for all mixer and all rotational speeds, there is a notably steeper increase at low size classes for both mixers when rotational speed is high. This increase can be mathematically described by a linear function fitted to the five smallest classes. If the slope of this function is viewed as an indicator for dispersive mixing, the results show that dispersive mixing is best for the Maddock mixing element at high screw speeds with the rhomboidal mixer at high screw speeds coming in second. For lower screw speeds the Maddock mixer shows superior performance compared to the rhomboidal mixer. This observation is congruent with established extrusion theory to a certain degree. The Maddock mixer delivering higher dispersive mixing performance is to be expected, but the fact that the rhomboidal mixing element at 80 kg/h surpasses the Maddock mixer at 20 kg/h is less intuitive. As mentioned above, the probable range of viscosities lies between 4.4 to 1 and 2.7 to 1, with the HDPE being the more viscous fraction. At this ratio of viscosity, breakup of drops of the dispersed phase is possible by both shear and elongation according to the theory of critical capillary numbers put forward by Grace [45]. As dispersive mixing by shearing is expected to be effective, yet less efficient than mixing by elongation, the authors’ findings are compatible with extrusion theory.

Considering the rather small amount of data present, the results described above do not reliably qualify that the method developed as suitable to analyse dispersive mixing. While the general trend expected from extrusion theory and physical background was met, further work is required to validate this trend, e.g., simultaneous analysis with other methods that assess dispersive mixing such as visual inspection of extruded films or atomic force microscopy. It should also be noted that (as already seen in temperature measurements seen in Section 3.1), the addition of low-viscosity polystyrene has a considerable influence on the build-up of shear and elongational forces within the melt. While it is very much possible to use the method developed to compare different mixing element designs, any absolute values (e.g., shear stress, elongational strain rate) extracted from extrusion trials conducted with the authors’ method must be expected to differ strongly from the values a geometrically identical mixer applies to an ‘undisturbed’ material during normal extrusion operations.

## 4. Discussion

This contribution presented an experimental setup meant to provide a high spatial resolution of the thermal, distributive and dispersive mixing process in dynamic mixing elements for single-screw extruders. This high spatial resolution was meant to improve the validation of mixing element simulation. In extrusion trials for a polyethylene main material at throughputs of 20 kg/h and 80 kg/h the effects of both a rhomboidal and a Maddock mixer as well as the effect of adding a black polyethylene and a black polystyrene melt upstream of the mixer were assessed.

It was found that multipoint temperature measurements upstream and downstream of the mixer are an efficient approach for validating the simulation compared to conventional single point temperature measurements. Even though single point measurements would have been sufficient to identify the considerable difference between simulated and experimental temperature profile that appear for one operating point, the added information about the general shape of the temperature profile will be useful in further developments of the simulation with the goal of eliminating the difference. Additionally, being able to measure temperature profiles upstream of a mixer instead of having to determine it in an iterative approach is likely to save labour, primarily in any research projects that need to consider the upstream temperature profile to determine the thermal mixing capacity.

In distributive mixing, an excellent spatial resolution of the colorant in the extrudate was observed. These high-fidelity experimental data allowed the confident determination of very good agreement between several simulations and experiment. In previous works, poor resolution of extrudate samples had prevented this. Using the experimental data alone, the rhomboidal mixer geometry was determined to require a modification that allows it to ‘scrape’ melt from the cylinder wall and move it towards the screw center. The authors see the great improvements made here as the key contribution of this work and hope to see it applied to numerous other situations in plastics processing requiring detailed flow analysis to improve e.g., mixing. However, the experimental setup so far has only considered mixing normal to the flow direction (‘cross mixing’). In practical extrusion applications, being able to provide a good level of ‘backmixing’, that is being able to smooth out temporal inhomogeneity in the melt, also is highly valued. Future work by the authors will consider this aspect of mixing by developing a variation of their setup that can be stopped and be disassembled without smearing colourant distribution, thereby allowing a practical validation of simulated residence time distributions. Another open question is whether the mechanisms of mixing remain the same when the size of melt inhomogenities decreases by orders of magnitude. To answer this question, coloured melt could be added upstream of the compression zone instead of upstream of the mixing element.

For dispersive mixing, a trend coherent with extrusion theory was found. As mentioned earlier, the addition of more experimental data and a more detailed analysis of the extrudates will prove whether the method shown here is sufficiently precise and unambiguous to be useful for mixing element design. At this point in time, analysis of dispersive mixing using the authors’ setup should only be used for very rough estimation of any mixer’s dispersive mixing capabilities. However, the potential to simultaneously analyse distributive and dispersive mixing by injecting a secondary stream of incompatible and differently coloured melt should not be disregarded.

## Figures and Tables

**Figure 1 polymers-12-02234-f001:**
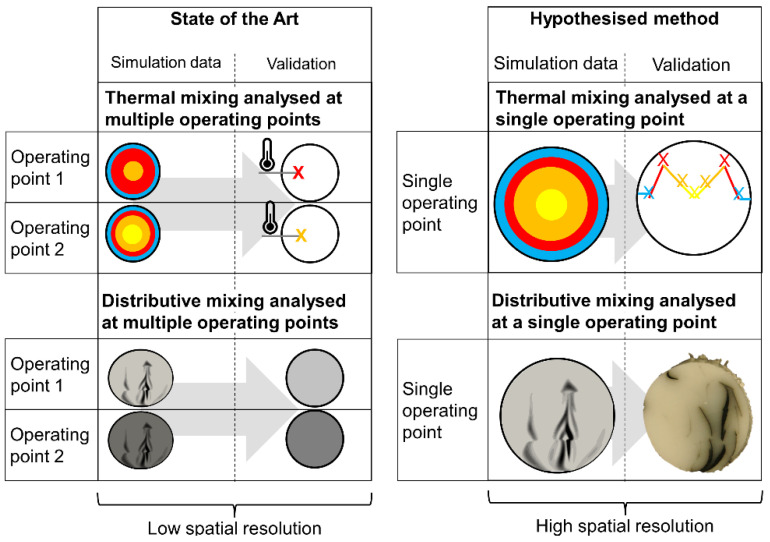
Validation of simulation of mixing elements in single-screw extrusion as done per state of the art (by comparing multiple operating points, with low spatial resolution) and as intended by the authors (at a single operating point with high spatial resolution).

**Figure 2 polymers-12-02234-f002:**
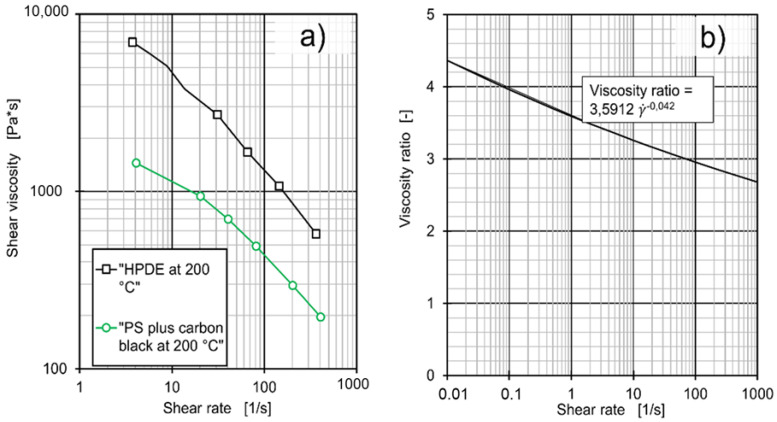
(**a**) Shear viscosity of Hostalen GD 9550 F [41] and PS 156F filled with carbon black as a function of shear rate, (**b**) viscosity ratio between the aforementioned melts as a function of shear rate.

**Figure 3 polymers-12-02234-f003:**
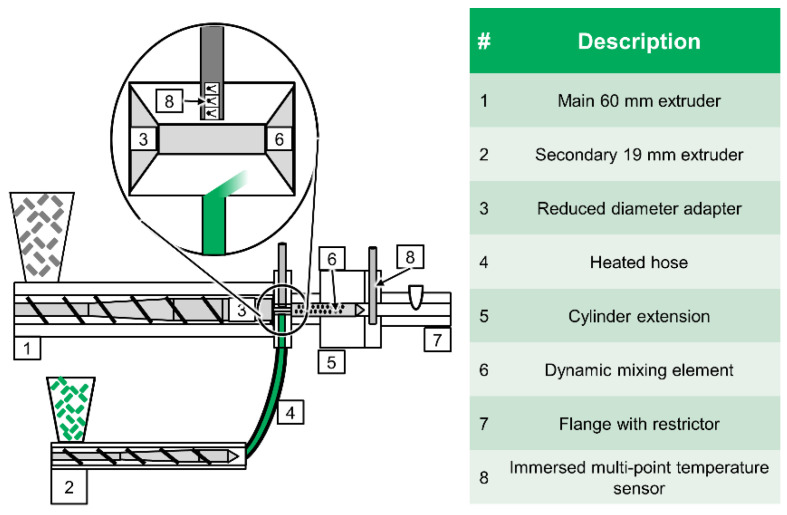
Schematic depiction of test setup for dynamic mixing elements for single-screw extruders.

**Figure 4 polymers-12-02234-f004:**
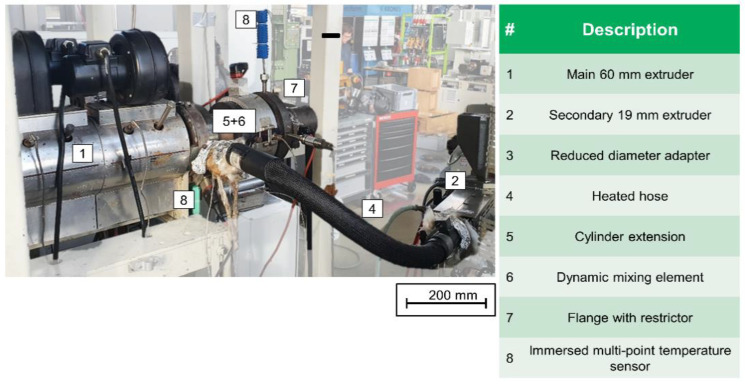
Photography of test setup for dynamic mixing elements for single-screw extruders.

**Figure 5 polymers-12-02234-f005:**
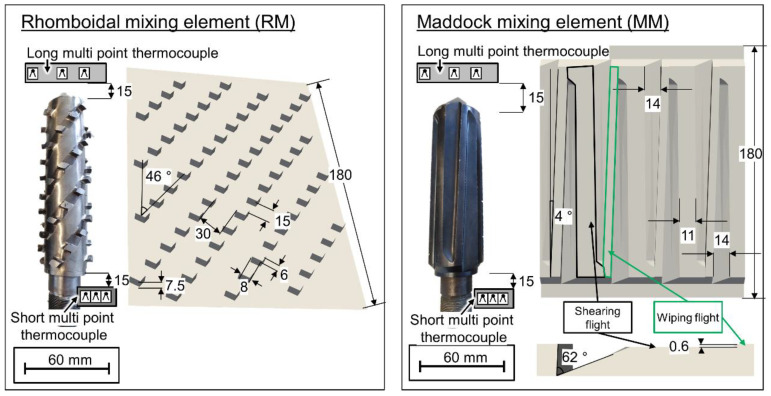
Photographic and schematic depiction of dynamic mixing elements used.

**Figure 6 polymers-12-02234-f006:**
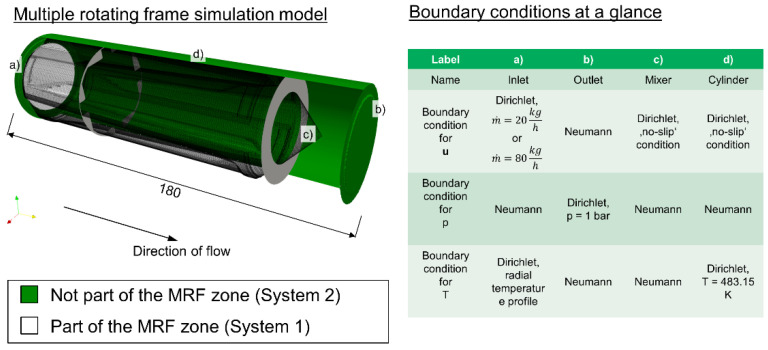
Graphical display for multiple reference frames (MRF) system setup and overview of boundary conditions.

**Figure 7 polymers-12-02234-f007:**
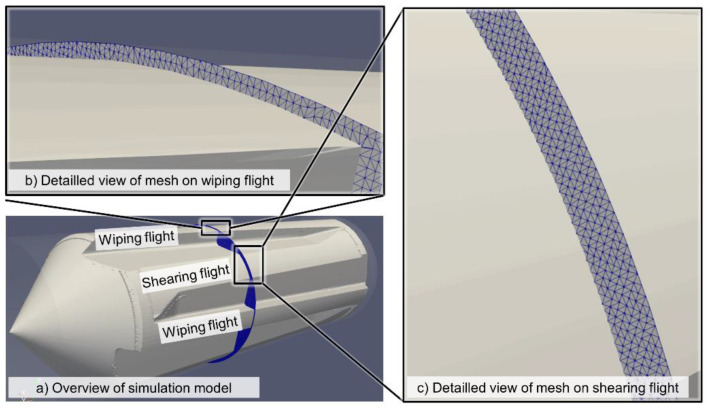
Detailed view of the Maddock mixer mesh, showing (**a**) the general layout, (**b**) the resolution of the gap between the wiping flight and the cylinder and (**c**) the resolution of the gap between the shearing flight and the cylinder.

**Figure 8 polymers-12-02234-f008:**
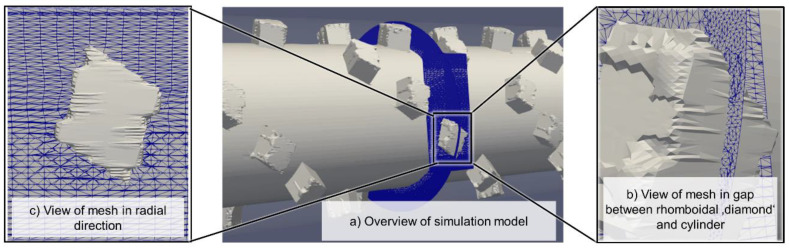
Detailed view of the rhomboidal mixer mesh, showing (**a**) the general layout, (**b**) the resolution of the radial gap between the top of the rhomboidal diamond and the cylinder and (**c**) the resolution of the mesh around the diamond as seen in the radial direction.

**Figure 9 polymers-12-02234-f009:**
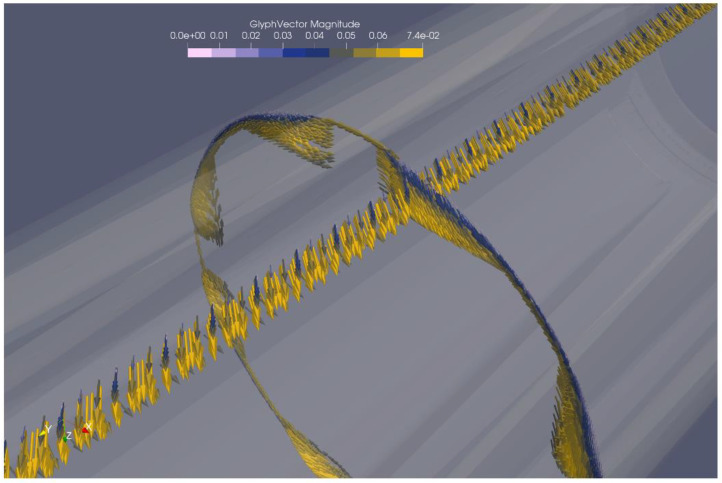
Flow velocity visualisation using vectorial arrows for the Maddock mixer.

**Figure 10 polymers-12-02234-f010:**
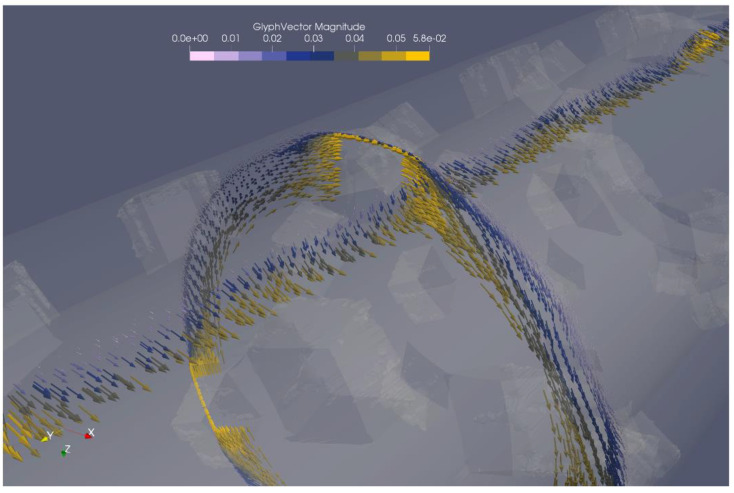
Flow velocity visualisation using vectorial arrows for the rhomboidal mixer.

**Figure 11 polymers-12-02234-f011:**
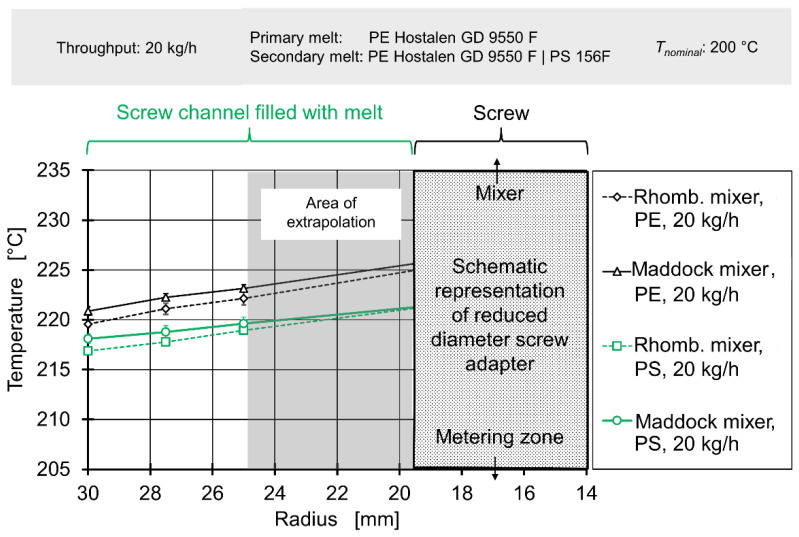
Radial temperature profile upstream of element for a throughput of 20 kg/h.

**Figure 12 polymers-12-02234-f012:**
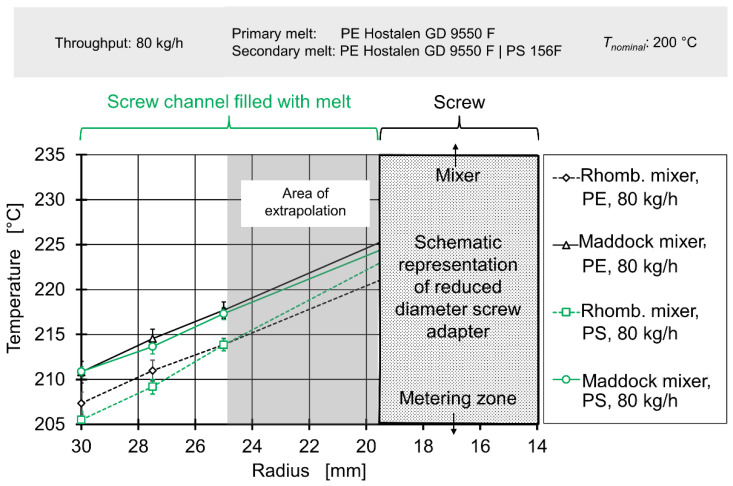
Radial temperature profile upstream of mixing element for a throughput of 80 kg/h.

**Figure 13 polymers-12-02234-f013:**
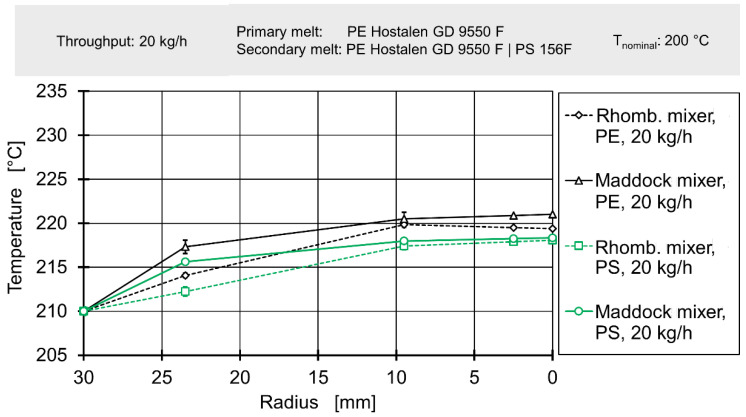
Radial temperature profile downstream of mixing element for a throughput of 20 kg/h.

**Figure 14 polymers-12-02234-f014:**
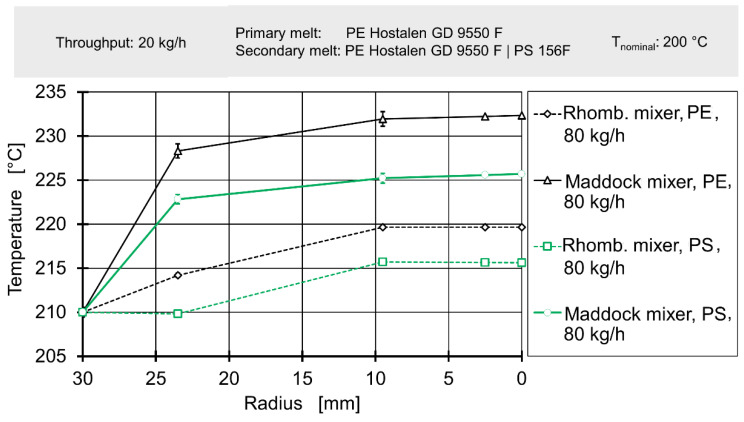
Radial temperature profile downstream of mixing element for a throughput of 80 kg/h.

**Figure 15 polymers-12-02234-f015:**
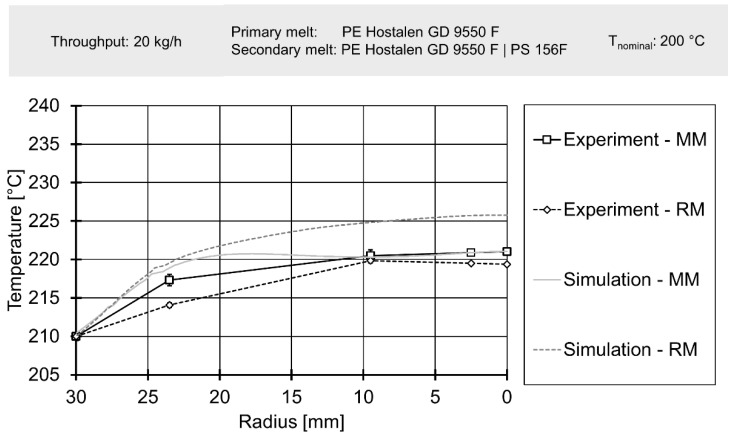
Radial temperature profile downstream of mixing element for a throughput of 20 kg/h.

**Figure 16 polymers-12-02234-f016:**
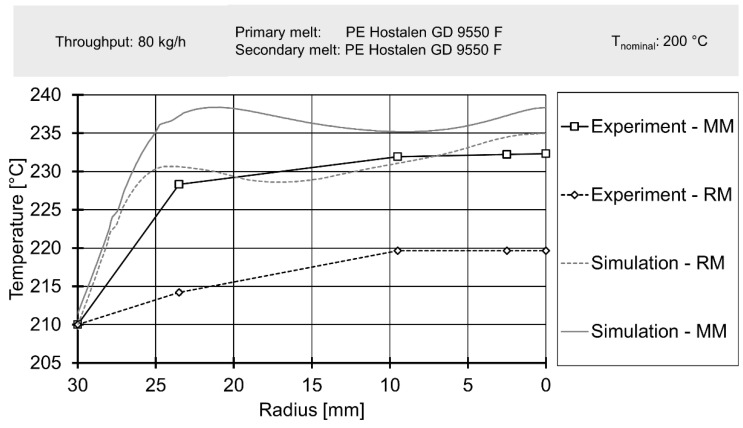
Radial temperature profile downstream of mixing element for a throughput of 80 kg/h.

**Figure 17 polymers-12-02234-f017:**
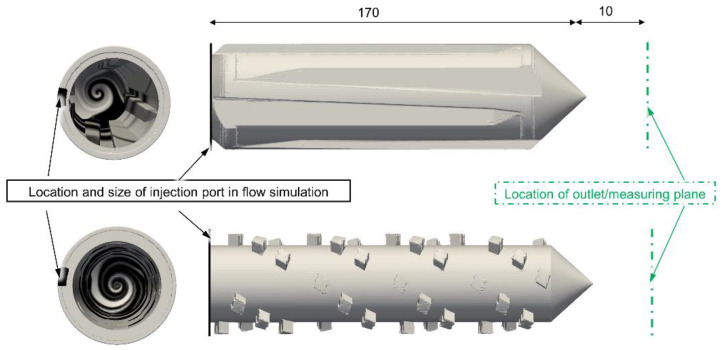
Location and size of injection port in flow simulation and location of outlet/measurement plane.

**Figure 18 polymers-12-02234-f018:**
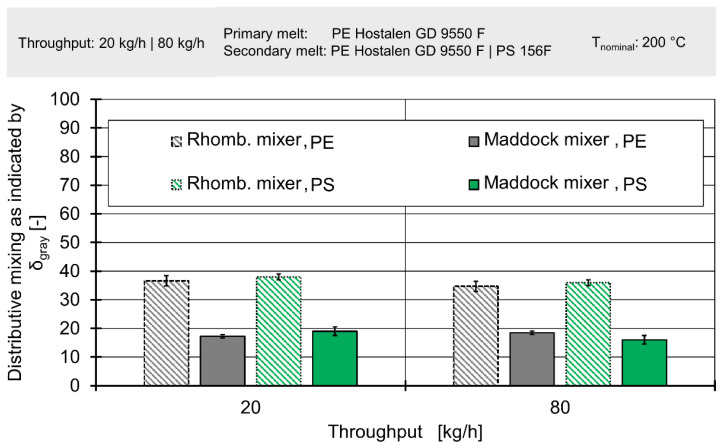
Experimental results for distributive mixing as expressed by δ_gray_ for different throughputs, different mixing elements and different secondary melts.

**Figure 19 polymers-12-02234-f019:**
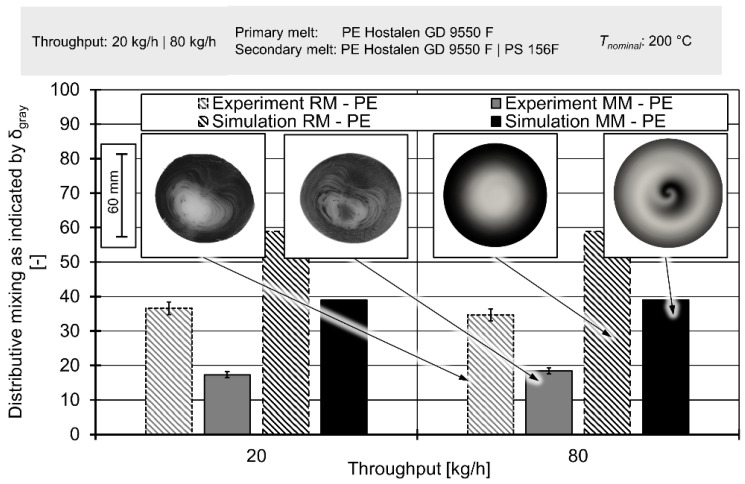
Distributive mixing in simulation and experiment as expressed by δ_gray_ for different throughputs and different mixing elements.

**Figure 20 polymers-12-02234-f020:**
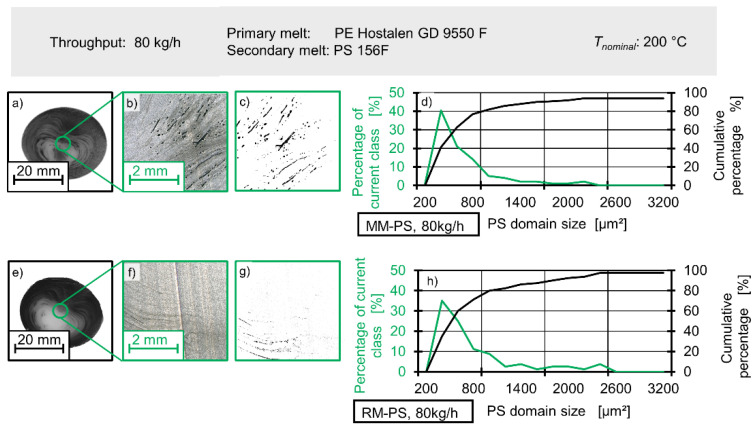
Analysis of dispersive mixing by bright field microscopy and evaluation of domain size.

**Figure 21 polymers-12-02234-f021:**
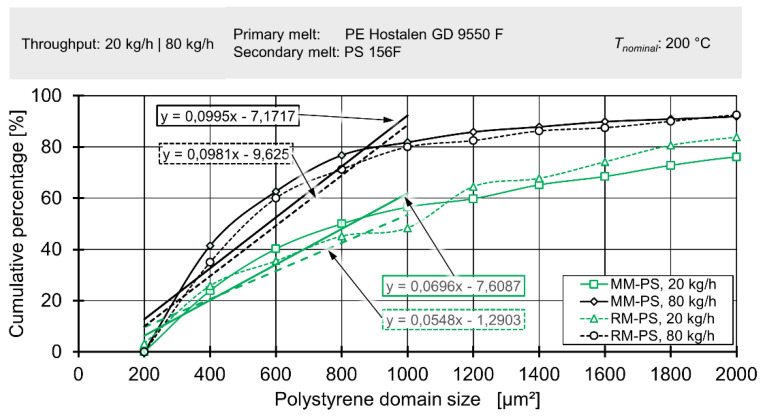
Distributive mixing in simulation and experiment as expressed by δ_gray_ for different throughputs and different mixing elements.

**Table 1 polymers-12-02234-t001:** Carreau parameters A, B and C for both high-density polyethylene (HDPE) and polystyrene (PS) with 20% carbon black.

Material	Carreau Parameter *A* [Pa * s]	Carreau Parameter *B* [1/s]	Carreau Parameter *C* [–]
HDPE	9472.83	0.19	0.66
PS plus carbon black	1735.32	0.08	0.61

**Table 2 polymers-12-02234-t002:** Zone-dependent cylinder temperatures for main extruder and secondary extruder.

Extruder	Feed Zone	Zones 1 and 2	Zones 3 and 4	Zones 5 and 6	Everything Else:
Main (60 mm)	20 °C	160 °C	180 °C	200 °C	210 °C
Secondary (19 mm), running PE	20 °C	160 °C	180 °C	200 °C	200 °C
Secondary (19 mm), running PS	45 °C	160 °C	180 °C	200 °C	200 °C

**Table 3 polymers-12-02234-t003:** Rotations per minute (RPM) for main extruder and secondary extruder.

Extruder	RPM at a Throughput of 20 kg/h	RPM at a Throughput of 80 kg/h
Main	18.9	88.2
Secondary (19 mm), running PE	20	80
Secondary (19 mm), running PS	20	80

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
