# Peer review of "A Method for the Validation of Simulated Mixing Characteristics of Two Dynamic Mixers in Single-Screw Extrusion"

_polymers, 2020, doi:10.3390/polym12102234_

Round 1

Author Response

Dear Sir or Madam,
thank you very much for your comments on our article „A method for the validation of simulated mixing characteristics of two dynamic mixers in single screw extrusion“ that has been submitted for review.

Please see the combined PDF file attached for the new version of the article and also a point-by-point overview of our responses to your comments.

Reviewer 2 Report

The work is interesting and has a potential for further practical directions. The investigations cover a large span of the numerical and experimental research. Unfortunately, there are so many defects in the manuscript:

  1. In numerical simulation part, the detailed description should be added to give a complete outline to show some key techniques to finish your trail. Such as some necessary descriptions of geometry model, boundary conditions and equations like WLF model. How about the material parameters? Equation 1 needs a schematic Figure to give a clear explanation. I guess you applied the rheology model of uniform distribution of PS for numerical simulation, which deviate from the true flow dynamics, some interpretations should be presented in terms of distributive mixing. How did you track the tracer mixing as shown in Figure 13? Please outline the technology procedure? The necessary numerical results should be provided, such as the velocity vectors, axial temperature contour and cross-section distribution, as well as pressure distribution contours.

  1. The experimental setup, mixer geometry, and test methodology should be improved to give a more clear and concise descriptions. The specific geometry like the injection pipe diameter and depth point should be given, which had an important effect on the mixing results. In Figure 5, please mark the temperature test positions upstream and downstream of the mixing section,

The outer diameter was 60 mm, so 60mm was encouraged to use instead of 75 mm. How did you collect the samples in time sequence? How many sample slice number did you deal with? From view of mixing dynamics, although your injection point was fixed, the colorant tracers were fed into the barrel from the different phase points with the result that they have different mixing dynamics, how do you treat these questions?

  1. As far as dispersive mixing, the viscosity ratio vs shear rate is an important factor. In Figure 2, the apparent viscosity of the black PS vs shear rate should be included to show the variation trend of viscosity ratio of PE to PS. X-axis should be 1000 rather than 1.000 in Figure 2.

The viscosity ratio vs the dominant shear rates by averaging the numerical simulation results should be discussed for the screw rotation speeds of 20 and 80RPM.

  1. There were some errors in grammars and words. In 121, rate should be rare? So many sentences are hard to follow, which should be polished once more.

  1. Many recent researches on chaos mixing with regard to simplified single screw model from Li-sheng Turng’s or XZ Zhu’s groups were not included, which will impose some important ideas on the understanding of mixing nature and initial sensitivity.

  1. There were some confusions in explaining Figure 1, more specific descriptions should be added.

Author Response

Dear Sir or Madam,
thank you very much for your comments on our article „A method for the validation of simulated mixing
characteristics of two dynamic mixers in single screw extrusion“ that has been submitted for review.

The pdf attached shows both the improved article and also - point-by-point- our responses to your comments.

Round 2

Reviewer 1 Report

The authors adressed all of my comments in full. I find the article suitable for publication in its present form.

Author Response

Dear Sir or Madam,

thank you very much for your time and your constructive criticism.

Reviewer 2 Report

The manuscript was improved to some extent. There are still some defects. Major revision is still needed.

  1. In numerical simulation section, the meshing details should be offered to see if the grid density is enough to identify the clearance regions and approach the rhomboidal geometry. Otherwise, the simulations do not make sense.

  1. The temperature distributions shown from Fig.7 to Fig. 11 seem strange, why the inner wall for screw element is higher than other radial positions? Is the dissipation responsible for this? If yes, perhaps the middle positions have chances. Please explain these Figures.

  1. Line193 to 250, expressions were not accurate. There should be the regions where the shear rate was lower than 15, even towards to zero where the vertex turns up. Some velocity vectors should be shown for two types of mixers for readers easy to follow. 

  1. Equation (4) is the expression of the balance of shear convection and the diffusion, independent of time evolution, which implies the infinite time result. Is this treatment correct? It is not clear where the cutting slices of mixed samples were taken? Actually, the coupled relationship between the tracer and matrix. Hence, the simplification was used and some factors were omitted. The apparent differences between the experimental and numerical results were found, please give some explanations.

  1. Line 323, ? The coordinate axes should be highlighted in Fig.6. Some statements should be revised accordingly. Some expressions is misleading when focusing on equation (3) and Line 323.

  1. In Fig.13, how about the axial location? Some denotes should be added to show where is where.

  1. Organization can be improved and the authors are encouraged to do this.

Author Response

Dear Sir or Madam,

thank you very much for your time and constructive criticism. You will find our point-by-point response to your comments attached as a PDF file.

Round 3

Reviewer 2 Report

The manuscript was improved to some extent. There are still some unclear details. Minor revision is still needed.

  1. For radical temperature distributions, there is another factor which should be included. The axial heat transfer in the screw element perhaps plays an important role. To my confusion, your description of the temperature conditions is not very clear. The axial cutting was recommended to show how temperature looks like for polymer melt and for the screw element itself. It should be noted that the steel’s heat transfer capacity is several hundred higher than that of polymer melt.
  1. Figures 11 and 12 are incorrect which are against the heat transfer law where the centerline point has higher temperature than the screw surface. Please get rid of the extrapolation line parts.
  1. The multi-point thermocouple positions were wrong for the downstream location, which did not agree with the upstream radical positions.
  1. In line 457, here should be Figures 14 and 15?
  1. In the introduction section, so many chapter X were used, please delete these lengthy expressions.

Author Response

Dear Sir or Madam,

thank you very much for your third round of comments on our article „A method for the validation of simulated mixing characteristics of two dynamic mixers in single screw extrusion“ that has been submitted for review.

I’d like to address these comments point-by-point in the attached PDF file. Changes made on account of your third round of comments are marked in violet on the new version of the article.
